# BRAIN-LIKE REPRESENTATIONAL STRAIGHTENING OF NATURAL MOVIES IN ROBUST FEEDFORWARD NEURAL NETWORKS

**Tahereh Toosi & Elias B. Issa**
Department of Neuroscience, Zuckerman Mind Brain Behavior Institute
Columbia University
New York, NY, USA
`{tahereh.toosi,elias.issa}`@columbia.edu

## ABSTRACT

Representational straightening refers to a decrease in curvature of visual feature representations of a sequence of frames taken from natural movies. Prior work established straightening in neural representations of the primate primary visual cortex (V1) and perceptual straightening in human behavior as a hallmark of biological vision in contrast to artificial feedforward neural networks which did not demonstrate this phenomenon as they were not explicitly optimized to produce temporally predictable movie representations. Here, we show robustness to noise in the input image can produce representational straightening in feedforward neural networks. Both adversarial training (AT) and base classifiers for Random Smoothing (RS) induced remarkably straightened feature codes. Demonstrating their utility within the domain of natural movies, these codes could be inverted to generate intervening movie frames by linear interpolation in the feature space even though they were not trained on these trajectories. Demonstrating their biological utility, we found that AT and RS training improved predictions of neural data in primate V1 over baseline models providing a parsimonious, bio-plausible mechanism – noise in the sensory input stages – for generating representations in early visual cortex. Finally, we compared the geometric properties of frame representations in these networks to better understand how they produced representations that mimicked the straightening phenomenon from biology. Overall, this work elucidating emergent properties of robust neural networks demonstrates that it is not necessary to utilize predictive objectives or train directly on natural movie statistics to achieve models supporting straightened movie representations similar to human perception that also predict V1 neural responses.

## 1 INTRODUCTION

In understanding the principles underlying biological vision, a longstanding debate in computational neuroscience is whether the brain is wired to predict the incoming sensory stimulus, most notably formalized in predictive coding (Rao & Ballard, 1999; Friston, 2009; Millidge et al., 2021), or whether neural circuitry is wired to recognize or discriminate among patterns formed on the sensory epithelium, popularly exemplified by discriminatively trained feedforward neural networks (DiCarlo et al., 2012; Tacchetti et al., 2018; Kubilius et al., 2018). Arguing for a role of prediction in vision, recent work found perceptual straightening of natural movie sequences in human visual perception (Hénaff et al., 2019). Such straightening is diagnostic of system whose representation could be linearly read out to perform prediction over time, and the idea of representational straightening resonates with machine learning efforts to create new types of models that achieve equivariant, linear codes for natural movie sequences. Discriminatively trained networks, however, lack any prediction over time in their supervision. It may not be surprising then that large-scale ANNs trained for classification produce representations that have almost no improvement in straightening relative to the input pixel space, while human observers clearly demonstrated perceptual straightening of natural movie sequences (subsequently also found in neurons of primary visual cortex, V1 (Hénaff et al.,

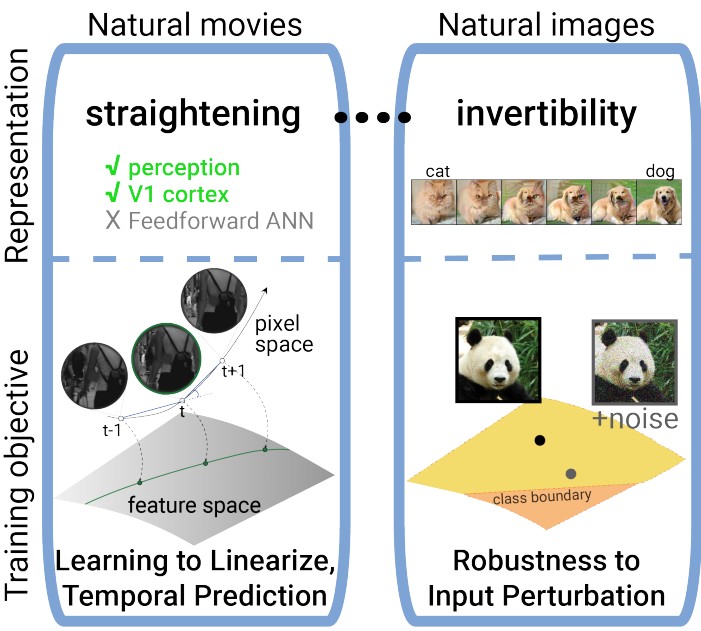

Figure 1: Perceptual straightening of movie frames can be viewed as invertibility of latent representations for static images. Left: straightening of representations refers to a decrease in the curvature of the trajectory in representation space such as a neural population in the brain or human perceptual space, but standard ANNs do not show straightening (Hénaff et al., 2019; 2021). Right: Invertibility of latent representation refers to interpolation between representation of two images (e.g. an image of a dog and an image of a cat), where the invertible interpolations show the main features of a dog morph into the main features of a cat. Invertible representations emerge in robust ANNs (Engstrom et al., 2019b), obviating the need to directly train for temporal straightening.

2019; 2021)). This deficiency in standard feedforward ANNs might suggest a need for new models trained on predictive loss functions rather than pure classification to emulate biological vision.

Here, we provide evidence for an alternative viewpoint, that biologically plausible straightening can be achieved in ANNs trained for robust discrimination, without resorting to a prediction objective or natural movies in training. Drawing on insights from emergent properties of adversarially-trained neural networks in producing linearly invertible latent representations, we highlight the link between perceptual straightening of natural movies to invertible latent representations learned from static images (Figure 1). We examine straightening in these robust feedforward ANNs finding that their properties relate to those in the biological vision framework. The contributions of this work are as follows:

1. We show that robust neural networks give rise to straightened feature representations for natural movies in their feature space, comparable to the straightening measured in the primate brain and human behavior, and completely absent from standard feedforward networks.

2. We show that linearly interpolating between the start and end frames of a movie in the output feature space of robust ANNs produces synthetic frames similar to those of the original natural movie sequence in image space. Such invertible linear interpolation is precisely the definition of a temporally predictive feature representation.

3. Compared to prior models of early visual cortex, robustness to input noise (corruption or adversarial robustness) is significantly better at explaining neural variance measured from V1 neurons than non-robustly trained baseline models, suggesting a new hitherto unconsidered mechanism for learning the representations in early cortical areas that achieves natural movie straightening.

## 2 RELATED WORK

### 2.1 MECHANISMS FOR PRODUCING BRAIN-LIKE REPRESENTATIONS

**Feedforward ANNs as models of biological vision.** Standard feedforward ANNs, although lacking a number of bio-plausible features such as feedback connections or a local learning rule (Whittington & Bogacz, 2019), still can explain the neural variance (Schrimpf et al., 2018) recorded from rodent (Bakhtiari et al., 2021), monkey (Yamins et al., 2014; Bashivan et al., 2019), and human visual cortex (Khaligh-Razavi & Kriegeskorte, 2014; Cichy et al., 2016) better than alternatives which are considered more bio-plausible by using a prediction objective function (e.g., PredNet and CPC (Zhuang et al., 2021; Schrimpf et al., 2020)). Thus, to learn the representations in the brain, regardless of bio-plausibility of mechanisms, feedforward ANNs provide a parsimonious more tractable class of leading models for object recognition in the visual cortex.

**Models of primary visual cortex.** In neuroscience, rather than rely solely a on top-down training objective like standard ANNs do, there has been a tradition of explaining early visual representations using more fundamental principles such as sparse coding and predictive coding as well as invoking unsupervised training (Olshausen & Field, 1996; Rao & Ballard, 1999). For example, unsupervised *slow feature analysis* extracts the slow-varying features from fast-varying signals in movies based on the intuition that most external salient events (such as objects) are persistent in time, and this idea can be used to explain the emergence of complex cells in V1 (Berkes & Wiskott, 2005). Recent work in machine learning has attempted to blend more bottom-up principles with top-down training by experimenting with swapping out ANN early layers with V1-like models whose filters are inspired from neuroscience studies (Dapello et al., 2020). This blended model turns out to have benefits for classification robustness in the outputs. However, it remains unclear whether there is a form of top-down training that can produce V1-like models. Such a mechanism would provide a fundamentally different alternative to prior proposals of creating a V1 through sparse coding or future prediction (Hénaff et al., 2019; 2021).

### 2.2 TEMPORAL PREDICTION AND INVERTIBILITY IN NEURAL NETWORKS

**Learning to predict over time.** Changes in architecture, training diet (movies), and objective (predicting future frames) have all been explored as mechanisms to produce more explicit equivariant representations of natural movies (Lotter et al., 2016; van den Oord et al., 2018). Directly related to the idea of straightening, penalizing the curvature of representations of frames was used in *Learning to linearize* (Goroshin et al., 2015) to learn straightened representations from unlabeled videos. This class of models does not need supervision which makes them more bio-plausible in nature; however, as mentioned in the previous section, they lag behind supervised feedforward ANNs both in terms of learning effective representations for object recognition and in producing feature representations that predict neural data.

**Learning invertible latents.** In deep learning applications, invertibility is mostly discussed in generative neural networks as a constraint to learn a prior to address applications in signals and systems such as image de-noising, signal compression and image reconstruction from few and noisy measurements or to be able to reconstruct or modify real images. Usually invertibility is implemented by carefully designing dedicated architectures (Jacobsen et al., 2018b; Chen et al., 2019). However, recently it has been shown it can be implemented in standard feedforward ANNs when they undergo training for adversarial robustness (Engstrom et al., 2019b;c). These works showed empirically that adversarially robust training encourages invertibility as linear interpolation between classes (e.g., cat to dog) results in semantically smooth image-to-image translation (Engstrom et al., 2019b) as opposed to blurry image sequences produced by standard ANNs.

We reasoned that robust networks which encourage invertibility may also lead to straightening as this a property that would be related to improved invertibility of a network, so we sought to extend prior work and study the behavior of robustly trained networks specifically in the domain of natural movies. We report on how these networks straighten natural movies in their features spaces and can invertibly reproduce movie frames in a natural sequence.

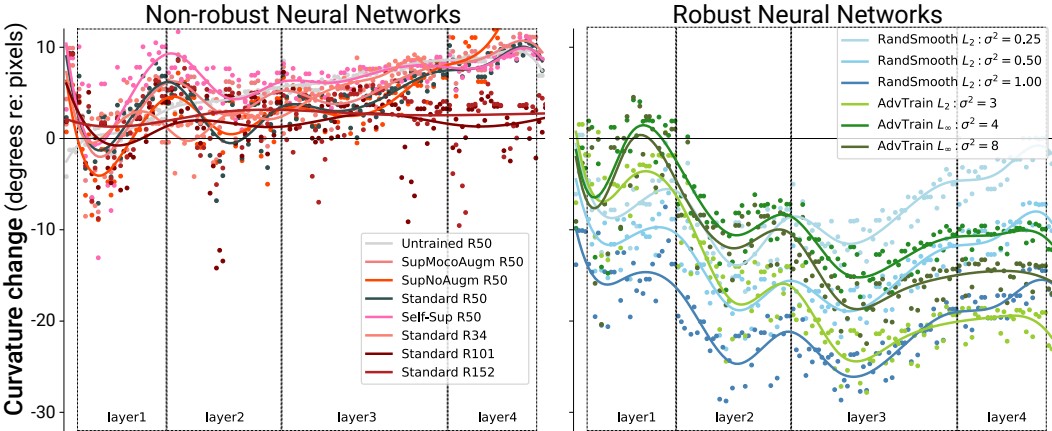

Figure 2: ANNs show straightening of representations when robustness to noise constraints (noise augmentation or adversarial attack) is added to their training. Measurements for straightening of movie sequences (from (Hénaff et al., 2019), in each layer of ResNet50 architecture under different training regimes: supervised training (standard), no training (random parameters), self-supervised training (Zbontar et al., 2021), supervised training with no augmentations, supervised training with extensive augmentations, supervised training with noise augmentation (base classifiers for RS) (Cohen et al., 2019), and supervised training with adversarial training (Engstrom et al., 2019a)

## 3 METHODS

### 3.1 BASELINE MODELS

We consider the class of feedforward convolutional neural networks, typically restricting to the ResNet-50 (He et al., 2015) architecture trained on ImageNet for the main analyses. Baseline networks (not trained for robustness) include a supervised ResNet-50/ResNet-101/ResNet-152, and self-supervised (Barlowtwins (Zbontar et al., 2021)). We trained ResNet-50 for imagenet classification without augmentations and with extensive augmentations (Chen et al., 2020), labeled as *SupNoAugm* and *SupMocoAugm*, respectively. We also consider Voneresnet (biological V1 front-end (Dapello et al., 2020)) and ResNet-50 trained as a base network for action recognition (Chen et al., 2021) but include these as separate examples in the Appendix since they use a modified architecture.

Table 1: Clean accuracy and robust (attack: $L_2, \epsilon = 0.1$) accuracy for the models used. Except for the custom models, all the other models were obtained from the repository of the references. Note that RS here refers to the base classifier in random smoothing without probabilistic inference.

| Models | Clean accuracy | Robust accuracy | Model reference |
|---|---|---|---|
| RN50 AT $L_2 : \epsilon = 3$ | 58.50 | 57.81 | (Engstrom et al., 2019a) |
| RN50 AT $L_\infty : \epsilon = 4$ | 62.80 | 61.40 | (Engstrom et al., 2019a) |
| RN50 AT $L_\infty : \epsilon = 8$ | 48.29 | 47.01 | (Engstrom et al., 2019a) |
| RN50 RS $L_2 : \epsilon = 0.25$ | 39.40 | 36.01 | (Cohen et al., 2019) |
| RN50 RS $L_2 : \epsilon = 0.5$ | 23.75 | 22.21 | (Cohen et al., 2019) |
| RN50 RS $L_2 : \epsilon = 1$ | 10.62 | 10.17 | (Cohen et al., 2019) |
| RN50 Standard | 75.43 | 52.32 | (He et al., 2015) |
| RN50 No augmentation | 64.35 | 28.13 | custom |
| RN50 Extensive augmentation | 75.27 | 53.08 | custom |
| RN50 Self-supervised | 70.18 | 41.73 | (Zbontar et al., 2021) |

## 3.2 Models trained for robustness

We consider two forms of models trained for minimizing a classification loss $\mathcal{L}_{ce}$ in the face of input perturbations $\delta \in \mathbb{R}^{h \times w \times c}$ subject to constraints on the overall magnitude of perturbations in the input space, where $x$, $y$, $\theta$ are the network input, output, and classifier parameters, respectively:

$$\mathcal{L}_{ce}(\theta, x + \delta, y) \tag{1}$$

In adversarially trained networks, projected gradient descent from the output space finds maximal directions of perturbation in the input space limited to length $\epsilon$, and training entails minimizing the effect of these perturbation directions on the network's output (Madry et al., 2018). In random smoothing (Lecuyer et al., 2018; Cohen et al., 2019), a supervised network is trained but in the face of Gaussian noise added to the input space as the base classifier before performing a probabilistic inference. In this work, we only use the representations as learned in base classifiers without the probabilistic inference. The perturbations in the base classifiers $\delta$ thus can follow:

$$\delta_{rand} \sim \mathcal{N}(0, \sigma^2 I), \qquad \delta_{adv} := \arg\max_{|\delta|_p \leq \epsilon} \mathcal{L}_{ce}(\theta, x + \delta, y) \tag{2}$$

These defenses to input noise have different motivations. Adversarial robustness provides defense against white box attacks whereas random smoothing is protecting against general image corruptions. However, prior work has suggested a connection between corruption robustness and adversarial robustness (Ford et al., 2019). Theoretically, random smoothing leads to certified robustness (Cohen et al., 2019) and trains a condition of invertible networks (Jacobsen et al., 2018a), while adversarial robustness has been shown empirically to lead to invertible latent representations in networks (Engstrom et al., 2019b).

## 3.3 Representational Metrics

***Representational straightening*** estimates the local curvature $c$ in a given representation $r$ of a sequence of images (natural or artificial) of length $N$, $C_{seq} : \{x_{t_1}, x_{t_2}, ..., x_{t_N}\}$ as the angle between vectors connecting nearby frames, and these local estimates are averaged over the entire movie sequence for the overall straightening in that representational trajectory (same as (Hénaff et al., 2019)):

$$c_t = \arccos\left(\frac{r_t - r_{t-1}}{\|r_t - r_{t-1}\|} \cdot \frac{r_{t+1} - r_t}{\|r_{t+1} - r_t\|}\right), \quad C_{seq} = \frac{1}{N} \sum_{t=1}^{N-1} c_t \tag{3}$$

Lower curvature (angle between neighboring vectors) indicates a straighter trajectory, and in the results we generally reference curvature values to the curvature in the input space (i.e., straightening relative to pixel space). This metric has been utilized in neuroscience showing that humans tend to represent nearby movie frames in a straightened manner relative to pixels (Hénaff et al., 2019). This curvature metric is also closely related to objectives used in efforts to train models with equivariance by linearizing natural transformations in the world as an alternative to standard networks trained for invariant object classification (Goroshin et al., 2015; Sabour et al., 2017).

***Expansion.*** We define the radius of a sequence of images from a movie clip as the radial size of the minimum covering hyper-sphere circumscribing all points representing the frames in $r$ (Gärtner, 1999). We use this measure to supplement geometrical characterization of a movie sequence in pixel space and in a model's representational spaces. Like representational straightening values, expansion values for models in the main text are referenced to the radius measured in pixel space or to the radius measure for the same layer in a baseline network by simply dividing by those references. We used mini-ball, a publicly available python package based on (Gärtner, 1999) to measure radius of the covering hyper-sphere.

## 4 Results

### 4.1 Robust ANNs exhibit representational straightening

With insights from connections to invertibility (see Figure 1), we hypothesized representational straightening of movie trajectories could be present in robustly trained neural networks. We took

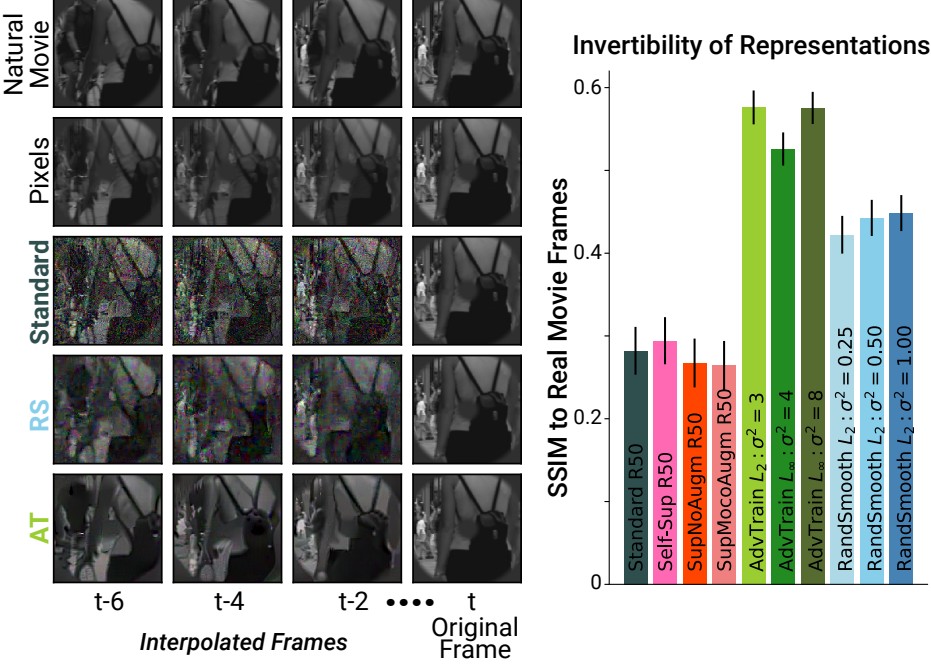

Figure 3: Invertibility as measured by the SSIM (Wang et al., 2004) of the actual in-between frames (labeled as Natural Movie) and the pixel-space projected linear interpolations between the first and the last frame labeled Pixels, standard ResNet50, RS (ResNet50, $L_2 : \sigma^2 = 0.5$) and AT (ResNet50, $L_2 : \sigma^2 = 3$). Interpolating the representations of the first and last frames in an invertible representation space generates a sequence of frames that are more similar to the ground-truth in-between frames, but for a non-invertible representation the generated frames are blurry and more similar to the interpolation in pixel space (a.k.a. artificial sequence). Interpolating the first and last frames in pixel space, second row, gives exactly what was called an artificial sequence in studies of straightening (Hénaff et al., 2019; 2021), as opposed to natural sequence which were the actual in-between frames.

the same movie stimuli publicly available (Hénaff et al., 2019)(A.4.1, Figure 12) and the same metrics, and we tested the same architecture, ResNet50 (He et al., 2015)) trained under different loss functions Table 1 to perform controlled head-to-head comparisons. Figure 2 shows representational straightening of natural movies measured in layers of ResNet50 trained under AT (Engstrom et al., 2019a) and RS (Cohen et al., 2019) at different adversarial attack or noise levels, respectively. Robust neural networks in contrast to other ANNs decreased the curvature of natural movies. Straightening for artificial sequences as measured in (Hénaff et al., 2019) (A.1, Figure 7) and other models (A.2, Figures 9 and 8) are provided in Appendix. Importantly, although most models, whether a standard ResNet-50 or one with a V1-like front-end, may display an initial dip in curvature for natural movies in the very earliest layers, this is not sustained in feature representations of later layers except for robustly trained networks (A.2, Figure 9 vs. A.1, Figure 7) and those trained on action recognition from temporally instructed training, which we include here as a proxy for a movie-like training though its feedforward architecture deviates from a ResNet50 by additional temporal processing components (A.2, Figure 8).

**Perceptual Straightening measured as invertibility of latent representations.** Next, we sought to empirically test how well robust networks can invert natural movies given that they contain linearized feature representation of movie frames in their high level feature spaces and given the general conceptual benefit of linearity for invertibility Figure 1. We measured invertibility of each model on the same movie sequences used for measuring straightening as follows. We linearly interpolated between latent representation of the first and last frame of each movie and used the same procedure as that used previously in (Engstrom et al., 2019b;a) to obtain the pixel-space correspondence of those interpolated representations. Whereas those generated pseudo-frames can be assessed by for their pixel-by-pixel distance to the actual movie frame, we chose a metric, *Structural Similarity In-*

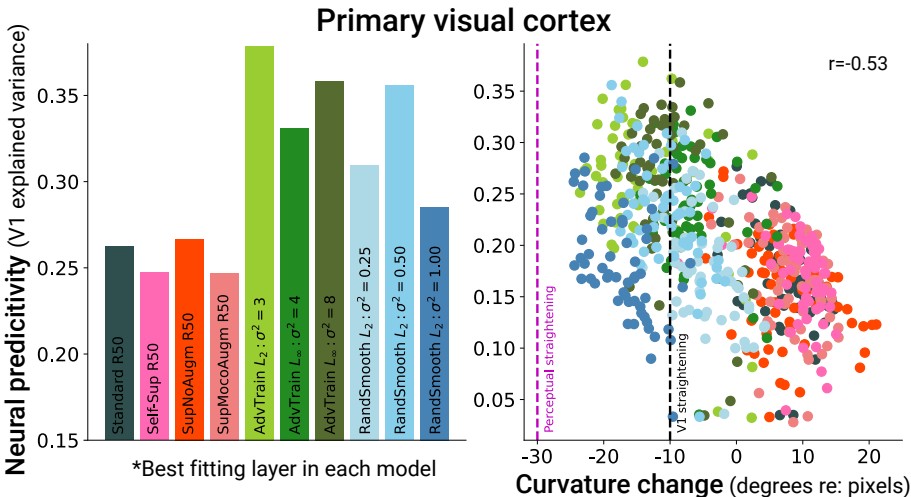

Figure 4: Left: RS and AT are more predictive of V1 neural responses than other non-robust models of the same architecture (ResNet50). Right: each dot represents a layer in ResNet50 trained under different loss function (color codes same as left). Higher representational straightening (negative curvature change) associates with higher V1 predictivity. Intriguingly, the highest V1 predictivity corresponds to layers that exhibit comparable straightening to that measured from V1 neurons ($-10°$ on average) (Hénaff et al., 2021). Explained variance is noise-corrected and computed as in (Schrimpf et al., 2018)

.

*dex Measure* (SSIM (Wang et al., 2004)), that utilizes intermediate-level statistics motivated from biological vision and putatively more related to some aspects of human perception than simple pixel space correspondence. Figure 3 shows an example of such inverted frames for standard ResNet50, RS ($L_2 : \sigma^2 = 0.5$) and AT ($L_2 : \sigma^2 = 3$), and a summary of average measured invertibility using the SSIM metric on pseudo-frames from each model. As expected, inline with the findings of previous work (Engstrom et al., 2019b), AT models scored relatively higher on invertibility of frames than a baseline discriminative model. However, what had not been previously shown is that RS models, using merely the benefits of their robustness to noisy augmentation (base classifier on top of learned representation; no probabilistic inference), also exhibit higher invertibility scores compared to standard trained models. Invertibility scores were consistently improved in RS and AT models across a variety of movies tested including those with relatively stationary textures and not just dynamic objects (see A.4.4, Figure 13 for further examples and A.4.3, Table 3 for scores across all 11 movies). Thus, RS models along with AT models exhibit invertibility of representations for movie frames which further demonstrates their ability to support perceptual straightening of natural movies in their highest layers that may be functionally similar to perceptual straightening previously measured from human subjects (Hénaff et al., 2019).

## 4.2 RANDOM SMOOTHING AND ADVERSARIAL TRAINING IN EXPLAINING NEURAL REPRESENTATIONS IN THE PRIMATE VISUAL SYSTEM

**Robustness to noise as a bio-plausible mechanism underlying straightening in primary visual cortex.** As shown above, straightening which is a constraint for brain-like representations in visual cortex manifests in robust neural networks. Both classes of RS and AT training for robustness to $L_2$ norm generate straightened representations of movie sequences. However, to distinguish among models of object recognition, we can measure how well they explain variance in patterns of neural activity elicited in different visual cortical areas. Here, for all neural comparisons in our analyses, we measured the Brain-Score (Schrimpf et al., 2018) using the publicly available online resource to assess the similarity to biological vision of each model, which is a battery of tests comparing models against previously collected data from the primate visual system (see Brain-Score.org). We found that RS and AT models provided a better model of V1 (in terms of explained variance) compared to non-robust models Figure 4. On other benchmarks, as we go up the ventral stream hierarchy from V1

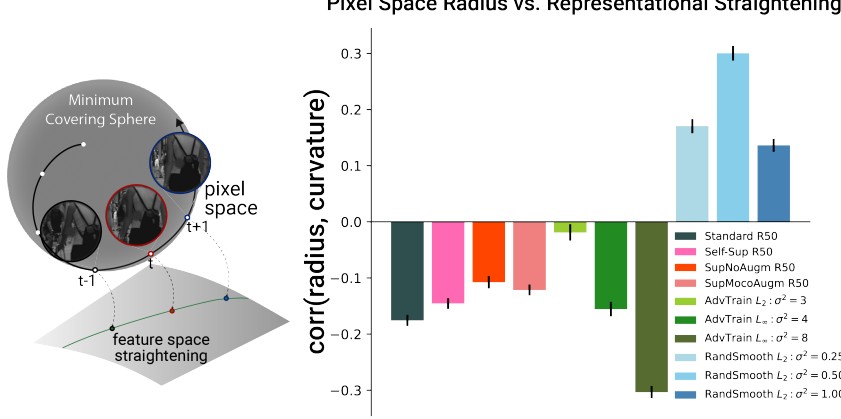

Figure 5: Can straightening for a movie sequence be explained by the size of the hyper-sphere bounding the frames (i.e. radius in pixel space)? While RS exhibits a small but positive correlation, the rest of the models, including AT, show negative or no correlations. Positive correlation means the smaller the size of the bounding hyper-sphere in pixel space, the more straightened the representation over the layers of the model.

to IT again, keeping the layer assignment fixed across models for proper comparison, we observed a decrease in explainability of robust models (A.3, Figure 11), in part presumably because robust models have lower object classification performance which is known to drive fits in higher brain areas like V4 and IT supporting object recognition (Yamins et al., 2014). Previous work (Dapello et al., 2020; Kong et al., 2022) linked adversarial robustness in models to their higher Brain-Score for V1, but we found that it may not be specifically driven by *adversarial* robustness per se, rather ($L_2$) noise robustness is also sufficient (as in base classifiers of RS tested here). More broadly, looking at neural fits across all models and their layers, we find that straightening in a particular model-layer correlates with improved explanatory power of variance in cortical area V1 (Figure 4, middle panel, each dot is a layer from a model), being even more strongly predictive than robustness of the overall model (A3, Figure 10). The level of straightening reached by best fitting layers of RS and AT models was comparable to the 10 degree straightening estimated in macaque V1 neural populations (black dashed reference line in Figure 4). This complements the fact that robust models peak near the 30 degree straightening measured in perception (Figure 2), suggesting that robust models can achieve a brain-like level of straightening to V1 and perception.

**Does the geometry of movie frame representations in pixel space dictate straightening in downstream representations?** The connection between two properties of the same representation manifold, robustness to independently sampled noise and straightened trajectories of smooth input temporal sequences, is not immediately clear. Because robustness is achieved by adding noise bounded by a norm ($L_2$, $L_2$, or $L_\infty$) in pixel space, a natural question is whether the radius of the bounding hyper-sphere of the frames of the tested movies in pixel space (see *Expansion* in Methods) was correlated with the measured straightening in feature space in each layer of the robustly trained models (Figure 5; also see A.5, Figure 14). We found, however, that there seemed to be different mechanisms at play for RS versus AT in terms of achieving straightening. RS models showed (small but) positive correlations, which means the smaller the ball containing all the frames of the movie in input space, the larger the straightening effect for the representations of frames of that movie in the model. While in AT models we see the opposite (negative) or no correlation. These divergent patterns underscore differences between these models and suggest that geometric size in pixel space is not strongly constraining the degree to which a movie can be straightened.

**Geometry of movie frame representations in feature space is relevant for capturing neural representations in V1** Between different RS models tested on different input noise levels, RS $L_2 : \sigma^2 = 0.5$ stands out as it gives a better model of V1 than those using smaller or larger magnitude input noise (Figure 4). For this model, we found that in addition to its intermediate level of straightening, the expansion score of movie frames, which is the radial size in its representation normalized to size in the same layer of a baseline ResNet50, was highest compared to the

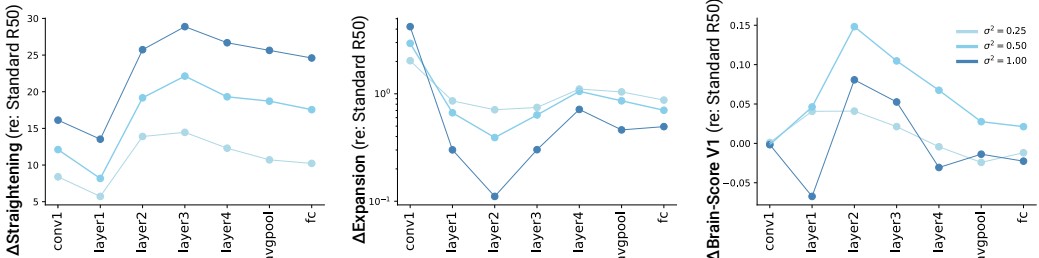

Figure 6: Geometric characteristics, straightening and curvature, of RS models related to V1 explainability. $\Delta$ means quantity is referenced to the same measure in a standard ResNet50.

other RS models (Figure 6, middle panel; measures are referenced to layers in a standard ResNet50 to highlight relative effect of robustness training rather than effects driven by hierarchical layer). This demonstrates a potential trade-off between improving straightening in a representation while avoiding too much added contraction of movies by robust training relative to standard training. This balance seems to be best achieved for $\sigma^2 = 0.5$, where we also see the significantly higher predictivity of V1 cortical data (Figure 6, right panel). The best AT model also shows little contraction of movies coupled with high straightening (A.5, 15).

## 5 DISCUSSION

We have demonstrated novel properties of robust neural networks in how they represent natural movies. Conceptually, this work establishes a seemingly surprising connection between disparate ideas, robust discriminative networks trained on static images on one hand, to work learning to linearize by training on natural movies, on the other. These modeling paths could both result in linearized, or straightened, natural movie representations (Figure 1). From a machine learning perspective, the invertibility and concomitant representational straightening of robust networks suggests that they learn explainable representations of natural movie statistics. Biologically, the emergence of straightening in these networks as well as their ability to better explain V1 data than baselines relatively lacking in straightening Figure 4 provides new insights into potential neural mechanisms for previously difficult to explain brain phenomena.

Biological constraints could lend parsimony to selecting among models, each with a different engineering goal. On face, RS by virtue of utilizing Gaussian noise instead of engineered noise gains traction over adversarial training as a more simple, and powerful way of achieving robustness in ANNs, which is inline with a long history of probabilistic inference in visual cortex of humans (Pouget et al., 2013). Indeed, looking across the range of robust models tested, the best fitting model of V1 was not necessarily the most robust but tended toward more straightened representations that also showed the least contracted representations – consistent with a known dimensionality expansion from the sensory periphery to V1 in the brain (Field, 1994). Future work exploring a wider variety of robustness training in conjunction with more bioplausible architectures, objectives, and training diets may yet elucidate the balance of factors contributing to biological vision.

At the same time, our work does not directly address how straightened representations in the visual system may or may not be utilized to influence downstream visual perception and behavior, and this connection is an important topic for future work. On the one hand, for supporting dynamical scene perception, behaviors that predict (extrapolate) or postdict (interpolate) scene properties over time (e.g., object position) may be supported by straightened natural movie representations. Indeed, both explanations, prediction and postdiction, have been invoked to account for psychophysical phenomena like the flash-lag illusion which present an interesting test case of how the brain processes complex stimuli over time (Eagleman & Sejnowski, 2000). However, even for relatively stationary scenes such as those containing textures, we observed benefits for straightening and invertibility in robustly trained networks (see A.4, Tables 2 and 3). Further work is needed to explore how spatially local versus global features in the presence of simple versus complex motion are affected in their relative straightening by model training.

ACKNOWLEDGMENTS

This work was supported by a Klingenstein-Simons fellowship, Sloan Foundation fellowship, and Grossman-Kavli Scholar Award as well as a NVIDIA GPU grant and was performed using the Columbia Zuckerman Axon GPU cluster. We thank all three reviewers for their constructive feedback that led to an improved final version of the paper.

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

# A APPENDIX

## A.1 STRAIGHTENING FOR BOTH NATURAL AND ARTIFICIAL SEQUENCES

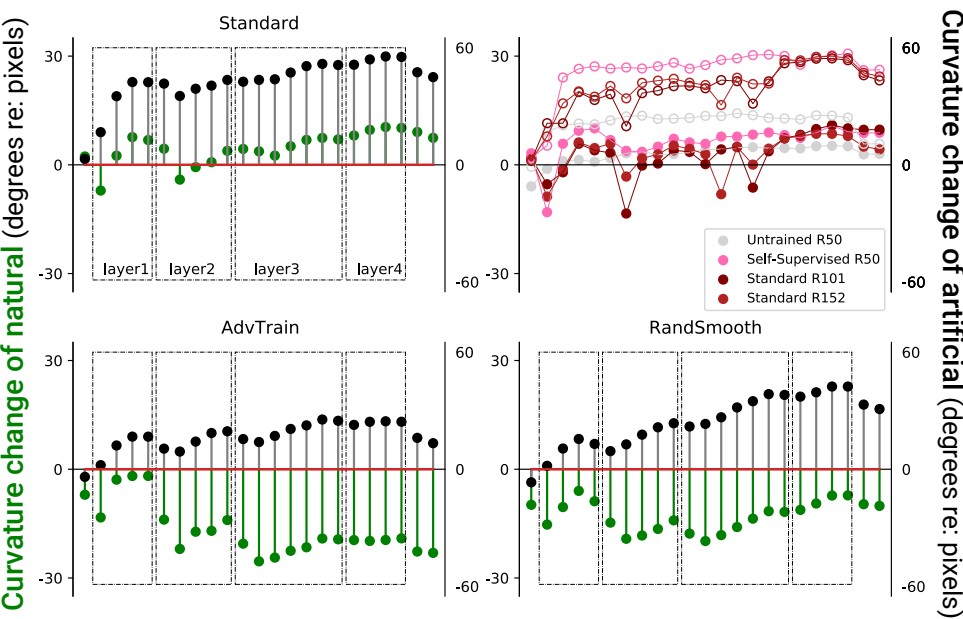

Figure 7: ANNs show straightening of representations when robustness to noise constraints (noise augmentation or robustness to adversarial attack) is added to their training. Counterclockwise from top left, measurements for straightening of movie sequences (from (Hénaff et al., 2019), natural sequence: green, artificial sequence: black) in each layer of ResNet50 architecture under different training regimes: supervised training (standard), supervised training with adversarial training ($L_2, \sigma^2 = 3$) (Engstrom et al., 2019a) and supervised training with noise augmentation ($L_2, \sigma^2 = 0.5$) (Cohen et al., 2019). Top right shows straightening for artificial (open circles) and natural (closed circles) sequences using ResNet architecture with no training (random parameters), self-supervised training (Chen et al., 2020) or additional layers.

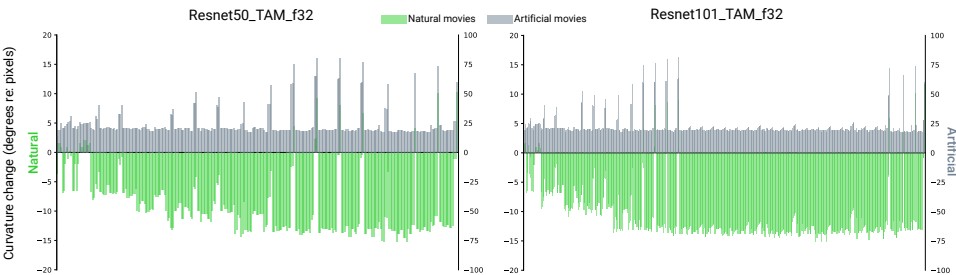

Figure 8: Straightening for ResNet50 and ResNet101 trained as the base architecture for action recognition (Chen et al., 2021) They were trained on video clips for action recognition. Although these models were not trained for straightening or predicting the next frame, they exhibit small but measurable straightening for natural movies. However, the curvatures for artificial sequences were not increased as much as curvature increase for artificial sequences in robust neural networks (Figure 7).

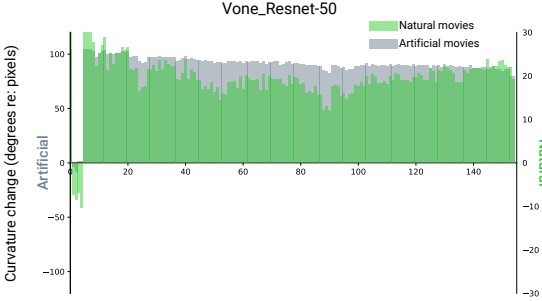

Figure 9: Lack of straightening for natural movies in ResNet50 trained with a biologically-inspired model of V1 in the front-end (Dapello et al., 2020). Vone-ResNet50 exhibits robustness to adversarial attack, but the fact that it does not exhibit straightening (except for the front-end) provides further evidence that adversarial robustness does not always accompany straightening.

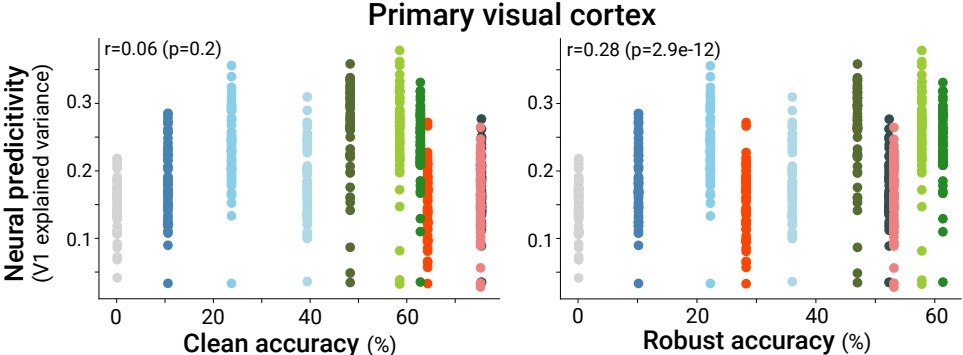

Figure 10: Clean accuracy (left) and robust accuracy (right) vs. V1 predictivity (same color convention as used in main text).

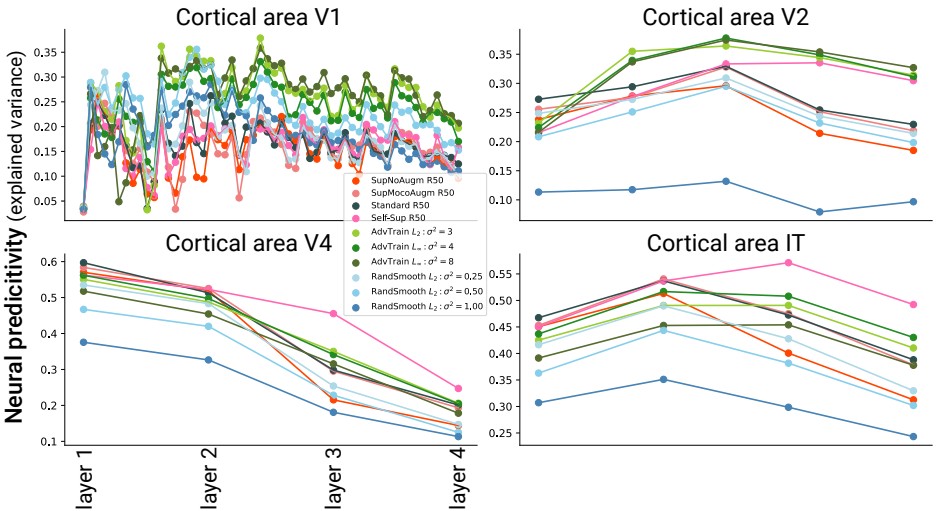

Figure 11: Brain-score (data and metric publicly available (Schrimpf et al., 2018)) for the models used in this study. The bar plot Figure 4 is a summary of this plot. Since the focus of this work was on V1 which is the only visual area for which neural straightening has been measured (Hénaff et al., 2021), we measured the brain-score for more layers for V1.

## A.4 MOVIE CHARACTERISTICS AND ADDITIONAL INTERPOLATION EXAMPLES

### A.4.1 MOVIES

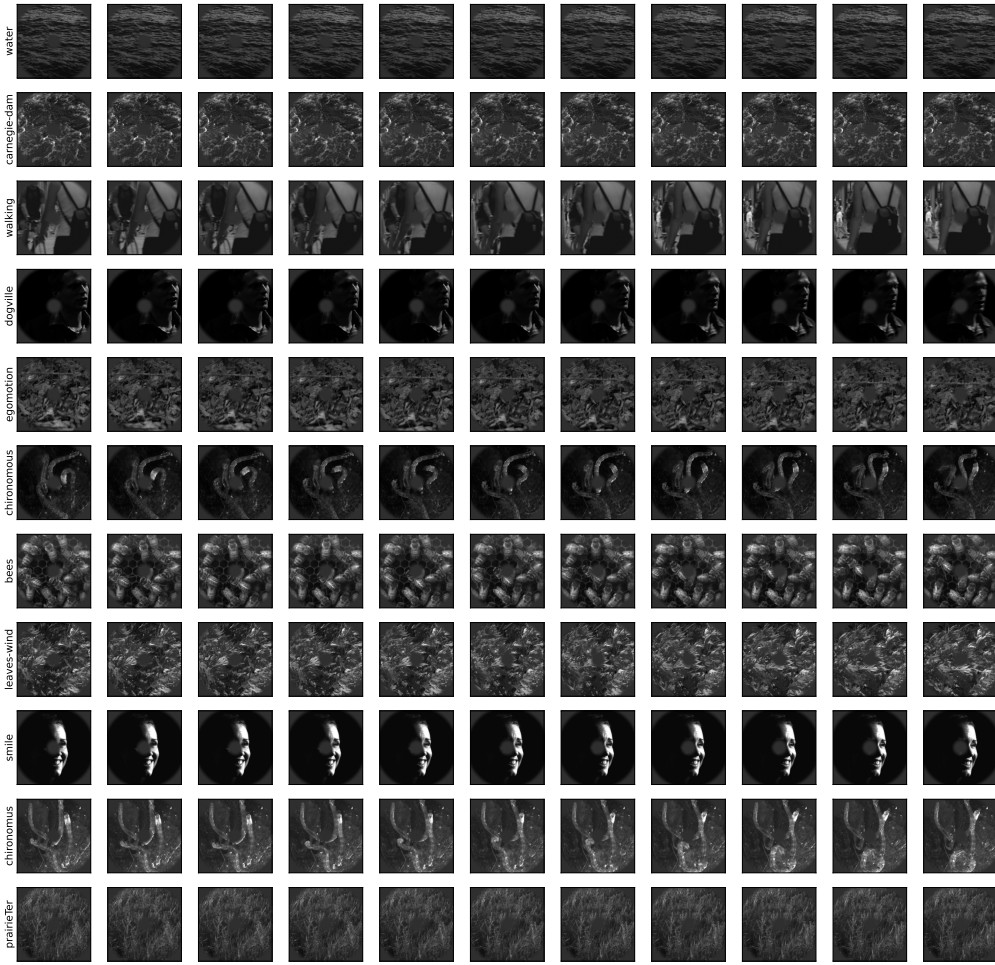

Figure 12: Movies used to evaluate straightening and invertibility (11 frames each).

## A.4.2 Table for average straightening

Table 2: Average curvature change (re: pixels) for each movie. RN stands for ResNet. The architecture of all robust models used was ResNet50.

|  | water | carn. | walk. | dogv. | egomo. | chiron. | bees | leaves | smile | chiron. | prair. |
|---|---|---|---|---|---|---|---|---|---|---|---|
| **RN18** | 45.29 | 36.25 | -27.58 | 12.39 | -34.82 | 13.30 | -5.94 | -58.03 | 9.28 | 0.69 | 48.45 |
| **RN34** | 44.79 | 36.63 | -28.43 | 12.39 | -35.67 | 12.39 | -5.81 | -58.65 | 12.08 | 0.43 | 48.14 |
| **RN50** | 48.85 | 37.33 | -28.13 | 14.00 | -34.68 | 13.45 | -5.66 | -58.53 | 10.79 | 0.76 | 49.72 |
| **RN50 Self-sup** | 52.05 | 40.32 | -26.73 | 18.98 | -33.19 | 14.37 | -5.68 | -58.22 | 19.42 | 0.52 | 49.90 |
| **RN50 MocoAugm** | 49.03 | 36.15 | -26.04 | 16.75 | -32.88 | 13.82 | -5.22 | -57.69 | 14.00 | 0.57 | 49.30 |
| **RN50 NoAugm** | 46.38 | 33.64 | -24.79 | 15.89 | -33.58 | 14.71 | -5.10 | -57.32 | 8.87 | 0.77 | 48.31 |
| **RN101** | 48.81 | 37.73 | -29.28 | 12.51 | -35.03 | 12.68 | -6.17 | -59.23 | 11.87 | -1.05 | 49.63 |
| **RN152** | 48.91 | 40.38 | -29.09 | 13.27 | -35.53 | 12.23 | -6.60 | -59.75 | 13.80 | -1.00 | 50.68 |
| **AT $L_2 : \epsilon = 3$** | 17.43 | 2.78 | -40.56 | -3.61 | -55.42 | -2.54 | -13.44 | -73.01 | -25.09 | -13.06 | 30.34 |
| **AT $L_\infty : \epsilon = 4$** | 25.93 | 16.51 | -36.55 | 3.48 | -50.46 | 5.45 | -12.90 | -68.74 | -11.49 | -7.91 | 38.46 |
| **AT $L_\infty : \epsilon = 8$** | 25.64 | 13.15 | -40.79 | 1.10 | -54.99 | 4.11 | -15.59 | -71.48 | -17.99 | -10.23 | 39.09 |
| **RS $L_2 : \epsilon = 0.25$** | 24.97 | 12.23 | -31.81 | 5.38 | -45.86 | 5.10 | -9.37 | -63.84 | -9.08 | -5.22 | 34.23 |
| **RS $L_2 : \epsilon = 0.5$** | 16.25 | 1.34 | -34.65 | 1.76 | -51.28 | 0.80 | -12.57 | -70.21 | -14.35 | -10.09 | 24.26 |
| **RS $L_2 : \epsilon = 1$** | 10.65 | -0.61 | -42.40 | -7.27 | -58.04 | -4.68 | -16.57 | -79.82 | -23.55 | -14.62 | 18.03 |

## A.4.3 Invertibility measure for each movie

Table 3: Average SSIMs for each movie. RN stands for ResNet. Architecture of all robust models used was ResNet50.

|  | water | carn. | walk. | dogv. | egomo. | chiron. | bees | leaves | smile | chirono. | prair. |
|---|---|---|---|---|---|---|---|---|---|---|---|
| **RN18** | 0.47 | 0.34 | 0.21 | 0.22 | 0.13 | 0.28 | 0.31 | 0.20 | 0.29 | 0.31 | 0.34 |
| **RN34** | 0.46 | 0.34 | 0.21 | 0.22 | 0.13 | 0.27 | 0.30 | 0.20 | 0.29 | 0.30 | 0.34 |
| **RN50** | 0.47 | 0.34 | 0.22 | 0.22 | 0.14 | 0.28 | 0.30 | 0.20 | 0.29 | 0.31 | 0.34 |
| **RN50 Self-sup** | 0.49 | 0.34 | 0.22 | 0.23 | 0.14 | 0.29 | 0.33 | 0.20 | 0.33 | 0.33 | 0.34 |
| **RN50 MocoAugm** | 0.44 | 0.32 | 0.20 | 0.21 | 0.12 | 0.25 | 0.27 | 0.19 | 0.27 | 0.28 | 0.31 |
| **RN50 NoAugm** | 0.43 | 0.32 | 0.20 | 0.21 | 0.12 | 0.26 | 0.32 | 0.19 | 0.28 | 0.28 | 0.32 |
| **RN101** | 0.48 | 0.35 | 0.22 | 0.23 | 0.14 | 0.28 | 0.31 | 0.21 | 0.30 | 0.30 | 0.35 |
| **RN152** | 0.49 | 0.36 | 0.22 | 0.23 | 0.14 | 0.29 | 0.31 | 0.21 | 0.30 | 0.31 | 0.36 |
| **AT $L_2 : \epsilon = 3$** | 0.76 | 0.59 | 0.52 | 0.67 | 0.22 | 0.62 | 0.58 | 0.35 | 0.81 | 0.63 | 0.59 |
| **AT $L_\infty : \epsilon = 4$** | 0.72 | 0.56 | 0.43 | 0.60 | 0.25 | 0.57 | 0.53 | 0.32 | 0.69 | 0.55 | 0.56 |
| **AT $L_\infty : \epsilon = 8$** | 0.78 | 0.59 | 0.45 | 0.66 | 0.29 | 0.65 | 0.56 | 0.35 | 0.78 | 0.61 | 0.62 |
| **RS $L_2 : \epsilon = 0.25$** | 0.53 | 0.48 | 0.35 | 0.47 | 0.18 | 0.44 | 0.44 | 0.29 | 0.57 | 0.46 | 0.42 |
| **RS $L_2 : \epsilon = 0.5$** | 0.60 | 0.49 | 0.37 | 0.51 | 0.22 | 0.44 | 0.45 | 0.30 | 0.54 | 0.49 | 0.46 |
| **RS $L_2 : \epsilon = 1$** | 0.61 | 0.50 | 0.35 | 0.53 | 0.24 | 0.44 | 0.48 | 0.30 | 0.52 | 0.49 | 0.48 |

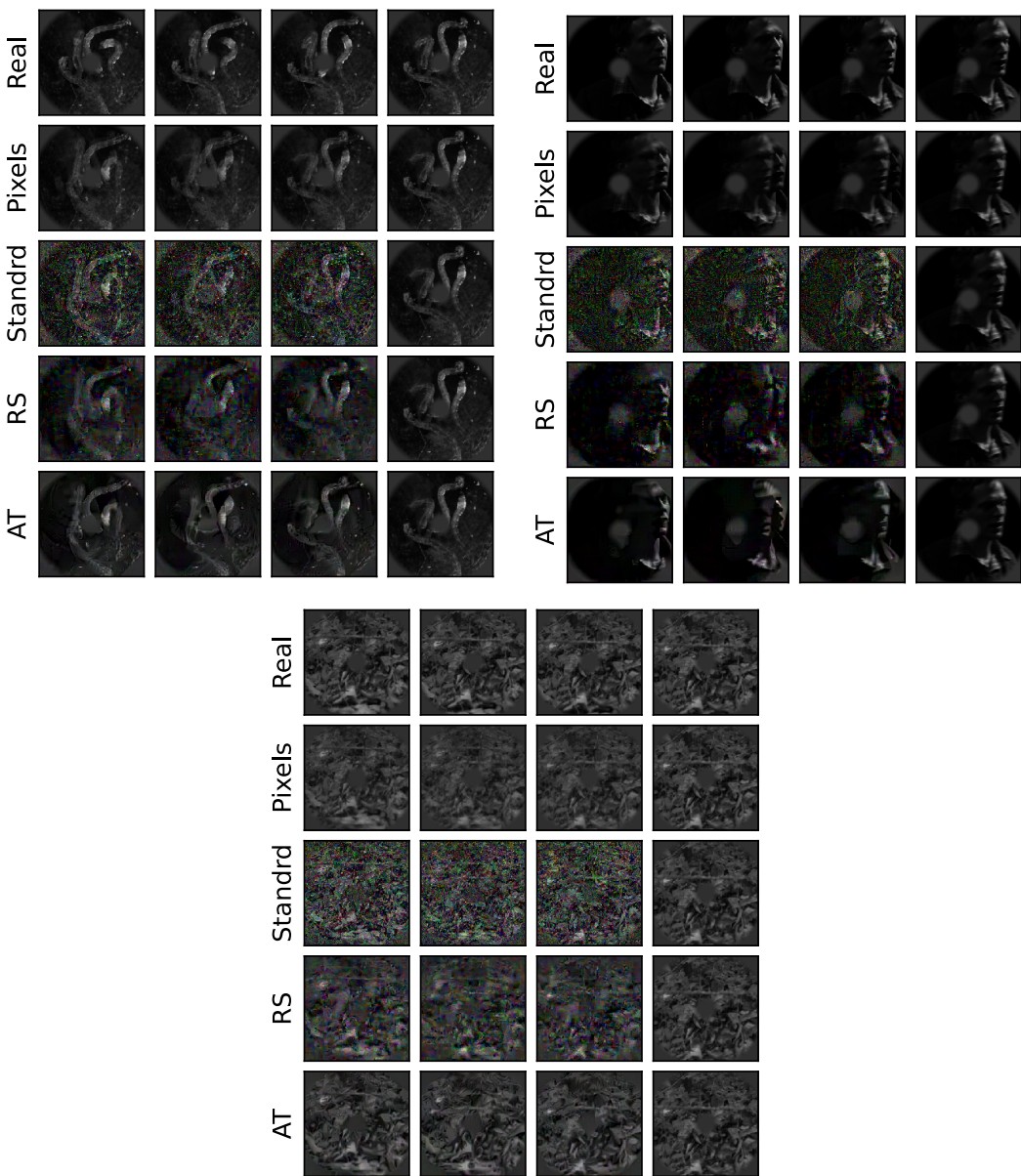

Figure 13: Three more example for interpolations for movies: chironomous, dogville, and egomotion, respectively. The gray dot in the middle of all frames is known as *fixation spot* where subjects (humans or monkeys) are instructed to keep their gaze toward during the experiment.

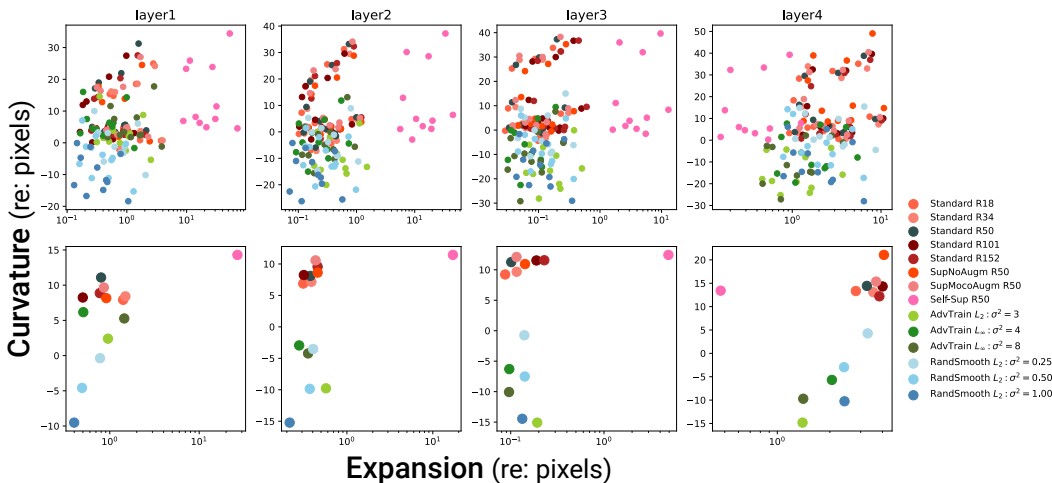

Figure 14: For each layer in each model, the expansions (re: pixels) and curvature change (re: pixels) were plotted for first row: all movies, second row: average over movies.

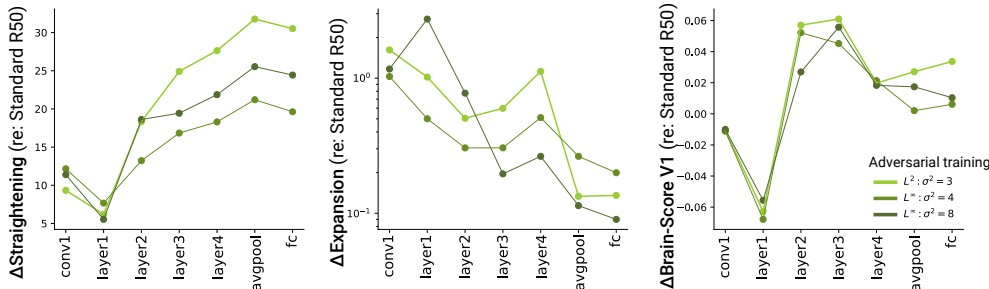

Figure 15: Geometric straightening and expansion of AT models and their relation to V1 explainability. Δ means quantity is referenced to the same measure in a standard ResNet50.

A.6    REPRODUCIBILITY INFORMATION

Almost all data (models, movies, and metrics) used in this work are publicly available and we provided references to them in the text (for instance see Table 1). We will release the code to reproduce the main results in this work at

`https://github.com/toosi/BrainLike_Straightening`

and we provide pointers to the publicly available resources used in this work as listed below.

**Movies and images.** We used the same movies used in the original studies on human perception and monkey primary visual cortex (Hénaff et al., 2019; 2021) which are available from first author Github as referenced in their papers. Images used to measure the clean accuracy and robust accuracy were taken from ImageNet validation set.

**Models.** All the models used in this study were from ResNet family and checkpoints for the main robust models are publicly available as references in the main text (Table 1). The checkpoints for the only two custom trained models (supervised with no augmentions and supervised with Moco augmentation) will be made publicly available along with the code.

**Neural predictivity metric.** We used brain-score, which is a publicly available benchmark to evaluate how well a model predicts variance in neural data (Schrimpf et al., 2018).

