# OpenReview forum: "Brain-like representational straightening of natural movies in robust feedforward neural networks"
_ICLR.cc/2023/Conference — ICLR 2023 poster_

### Official Review · Reviewer_JB2C · 2022-10-14

**Confidence:** 4
**Correctness:** 4
**Technical Novelty And Significance:** 1
**Empirical Novelty And Significance:** 3
**Recommendation:** 8

**Clarity, Quality, Novelty And Reproducibility:**

## **Clarity:**
Overall, the paper uses a consistent color scheme which is very helpful in understanding and relating the different figures. Especially Figure 2 is really cool and very clear.

**Major:** Suggesting to tone down language regarding "brain-like" representations. None of the plots show the ceiling for e.g. predicting neural data, so a relative comparison between computational models (e.g. model A better fits neural data than model B) must not be mistaken with stating that model A is brain-like / has a brain-like representation. I would suggest to give the paper a careful read and whenever the term "brain-like" is used, to either contextualize - e.g. stating that model A is *more* brain-like than model B - or to actually back the statement up by showing that the model is on par with neural systems. For instance, the title states "brain-like representational straightening", but this claim is never investigated in the paper and would involve comparing to the estimated neural curvature of neural systems when viewing exactly the same stimuli.

**Other points:**
- references throughout the paper are cited in the wrong citation style. If the sentence does not refer to the authors specifically as in Authors (2022), then the citation should be in brackets (Authors, 2022). Depending on the package, this can be achieved via \cite{} vs. \citep{} etc. Currently sentences with citations are often hard to read.
- overall sloppiness & typos, e.g. "BarlowtwinsZbontar", "front-endDapello", often related to the above point; also Figure 4 caption "is more brain-like representations"
- contributions point (1.) is descriptive without answering what the contribution is, the use of past tense ("gave rise") is suggestive of this being known from prior work whereas what the authors likely intend to convey is that this is their first contribution. Suggesting to re-phrase it as e.g. "We show that ..."
- Table 1 formatting is different from template (double columns on left/right)
- Figure 3 left: it's hard to see anything in the movie frames, I would suggest to use an example where one can see more. Currently I have a hard time understanding what's happening even in the natural movie row.
- Figure 3: cutting the y axis (instead of starting at zero) is misleading and contradicts good practice in data visualization. I would strongly suggest to start the y axis from zero so that the bar height is proportional to the effect.
- "EXPLAINING BRAIN-LIKE REPRESENTATIONS" (section heading): that's a very broad goal, can the authors be a bit more specific here in terms of which definition of "explaining" they are using? Is "explanation" really the best term, or would "fitting neural data" be more suitable perhaps? Different people have very different expectations when thinking of what an explanation would consistute.
- minor: page 8 "confirmed in a recent study Guo et al. (2022)" -> perhaps "as investigated in a recent study by Guo et al. (2022)?" I don't think the study answered the question to the degree that one would like to think of it as completely confirmed.
- why do the curves in Figure 2 not start from zero curvature difference for the input space (before layer 1, i.e. at x=0)?
- term "neural behaviour": can you clarify? I thought data is usually either neural data or behavioural data
- Given that the action recognition network and VOneResNet were introduced in the methods section, why are they not shown in the main figures of the paper?


## **Quality:**
**Major:** Figure 4 claims that RS fits brain data better than adversarial training: do the baseline ImageNet accuracies of the RS and the adversarially trained models match? If not, can you provide an accuracy-matched comparison? Given that the ImageNet accuracy of the chosen adversarially trained models is very low, it is not surprising that they're also bad at fitting brain data, so this would be a caveat in the comparision.

**Minor:** Movie interpolation example (Figure 3 left): I would suggest to show more examples in the appendix, ideally e.g. 10 examples that are randomly selected (rather than chosen by the authors).


## **Novelty:**
To the best of my knowledge, establishing a connection between adversarial training and perceptual straightening (Figure 2) is a novel and important contribution.

The investigation of the invertability is less novel, as the authors mention this is known from robust models and the RS model performs worse than the adversarially trained models in this regard.

In terms of the neural fits (e.g. Figure 4), it would help to explain where data is simply taken from brain-score.net and where it is computed/evaluated by the authors.

## **Reproducibility**:
Very poor - no code submitted, not enough details. For instance: How is the RS model trained? What is its ImageNet accuracy? Which hyperparameters, data augmentation, or even which deep learning library were used during training? I would strongly suggest to improve the reproducibility by providing more details in the paper itself, alongside submitting code if at all possible, and if this is not possible to explain why.

**Strength And Weaknesses:**

Strengths:
+ establishes a connection between adversarial training/random smoothing and perceptual straightening, which is novel and very interesting
+ consistent color scheme, (mostly) clear figures, especially Figure 2 conveys the main point of the paper

Weaknesses:
- Brain-score comparison may be impacted by different baseline accuracies between adversarially trained and RS models (had to tell, since I couldn't find the RS model's baseline accuracy in the paper)
- some claims / wording not backed up by data ("brain-like representational straightening")
- multiple issues regarding clarity, see suggestions below
- poor reproducibility


**Summary Of The Paper:**

[response to author's rebuttal: see https://openreview.net/forum?id=mCmerkTCG2S&noteId=mpYhm0LBTQD]

The paper "Brain-like representational straightening of natural movies in robust feedforward neural networks" investigates to which degree standard ImageNet-trained models and "robust" networks (in terms of adversarial training, training with random smoothing) straighten representational trajectories of short movie sequences, which is an effect that is known from neuroscience. They find that robust networks indeed straighten representational trajectories. A few other experiments investigate links to fitting brain data and inverting representations.

**Summary Of The Review:**

I am leaning towards acceptance since investigating the link between adversarial training / noise robustness and representational straightening is a clear, novel and important finding. At the same time, the paper currently has a number of issues that should be improved (see suggestions above). I would be happy to raise my score if my concerns are sufficiently addressed.

---

> ### Author Response · Authors · 2022-11-19
> **Initial response to review by JB2C (Part 1 of 3)**
>
> We would like to thank the reviewer for their helpful feedback and constructive suggestions to improve the manuscript. We attempt to address each of these below and hope that these changes improve the clarity and quality of the manuscript.
>
> >**CLARITY (MAJOR) \
> Suggesting to tone down language regarding "brain-like" representations…I would suggest to give the paper a careful read and whenever the term "brain-like" is used, to either contextualize - e.g. stating that model A is more brain-like than model B - or to actually back the statement up by showing that the model is on par with neural systems. For instance, the title states "brain-like representational straightening", but this claim is never investigated in the paper and would involve comparing to the estimated neural curvature of neural systems when viewing exactly the same stimuli.**
>
> Thank you very much for bringing this to our attention. We agree that the term brain-like should be used with care and done in the appropriate context. In this context, our intended usage of brain-like should be in reference to the observed representational straightening in models being comparable to humans. Indeed, prior work measured straightening for these very same movies in human behavior and the macaque primary visual cortex providing a basis for saying “brain-like representational straightening” in models. Section 4.2, Figure 4B presents this connection to prior brain measurements by putting reference lines in the scatter plot of Figure 4B and stating:
>
> “The level of straightening reached by best fitting layers of RS and AT models was comparable to the 10 degree straightening estimated in macaque V1 neural populations (black dashed reference line in Figure 4). This complements the fact that robust models peak near the 30 degree straightening measured in perception (Figure 2), suggesting that robust models can achieve a brain-like level of straightening to V1 and perception.”
>
> As such, in the rest of the manuscript we have removed references to brain-like that are outside of this context and used more specific language. For example, the title for Section 4.2 now reads “in explaining neural representations in the primate visual system” instead of “in explaining brain-like representations.”
>
> However, we retain “Brain-like representational straightening” in the title specifically because that motivates examining straightening and we provide a comparison of model straightening levels to straightening as measured in neuroscience experiments.

---

> > ### Author Response · Authors · 2022-11-19
> > **Initial response to review by JB2C (part 2 of 3)**
> >
> > >**CLARITY (MINOR)**
> >
> > >* references throughout the paper are cited in the wrong citation style…Currently sentences with citations are often hard to read.
> >
> > We have fixed all citation formatting in the manuscript. We apologize for any inconvenience in the reading of the initial manuscript.
> >
> > >* overall sloppiness & typos, e.g. "BarlowtwinsZbontar", "front-endDapello", often related to the above point; also Figure 4 caption "is more brain-like representations"
> >
> > We have attempted to clean up typos and figure labels.
> >
> > The caption for Figure 4 no longer contains the term “brain-like” and instead presents a new figure showing reference lines for straightening measured in macaque cortex and human perception to biologically ground curvature measured in models.
> >
> > >* contributions point (1.) is descriptive without answering what the contribution is, the use of past tense ("gave rise") is suggestive of this being known from prior work whereas what the authors likely intend to convey is that this is their first contribution. Suggesting to re-phrase it as e.g. "We show that ..."
> >
> > Thanks for this suggested clarification. We have switched to the suggested phrasing.
> >
> > >* Table 1 formatting is different from template (double columns on left/right)
> >
> > Table 1’s formatting has been fixed.
> >
> > >* Figure 3 left: it's hard to see anything in the movie frames, I would suggest to use an example where one can see more. Currently I have a hard time understanding what's happening even in the natural movie row.
> >
> > We had attempted to choose a movie with some interpretable motion. Unfortunately, the ~1 second long movie clips used in prior work have fairly limited motion partly for experimental constraint reasons. To aid the reader, we have included frames from all the movies in Appendix A.4.1 as well as more examples of the interpolations from models in A.4.4. Short of a perfect way to display many frames, we suggest digitally zooming in which does reveal more detail as the images are embedded at high resolution in the electronic copy.
> >
> > >* Figure 3: cutting the y axis (instead of starting at zero) is misleading and contradicts good practice in data visualization. I would strongly suggest to start the y axis from zero so that the bar height is proportional to the effect.
> >
> > Fair point. Figure 3 y-axis now starts from zero.
> >
> > >* "EXPLAINING BRAIN-LIKE REPRESENTATIONS" (section heading): that's a very broad goal, can the authors be a bit more specific here in terms of which definition of "explaining" they are using? Is "explanation" really the best term, or would "fitting neural data" be more suitable perhaps?
> >
> > Agreed, the title now reads “explaining neural representations in the primate visual system.”
> >
> > >* minor: page 8 "confirmed in a recent study Guo et al. (2022)" -> perhaps "as investigated in a recent study by Guo et al. (2022)?" I don't think the study answered the question to the degree that one would like to think of it as completely confirmed.
> >
> > Upon reconsidering the sentence containing the Guo et al. (2022) reference, we have decided to remove the sentence since it’s not a well-supported statement which was too speculative – as the reviewer points out, it has not been completely confirmed.
> >
> > >* why do the curves in Figure 2 not start from zero curvature difference for the input space (before layer 1, i.e. at x=0)?
> >
> > In ResNet50 before layer 1 there are modules (conv1, bn1, maxpool), layer 1 refers to the first residual layer. We stick to the common naming convention for consistency.
> >
> > >* term "neural behaviour": can you clarify? I thought data is usually either neural data or behavioural data
> >
> > We believe that the reviewer is referring to the last sentence in the Abstract. Instead of saying “that predict neural behavior,” we now specifically state “that also predict V1 neural responses”. Here is the updated last sentence of the Abstract:
> >
> > “to achieve models supporting straightened movie representations similar to human perception that also predict V1 neural responses.”
> >
> > >* Given that the action recognition network and VOneResNet were introduced in the methods section, why are they not shown in the main figures of the paper?
> >
> > We chose to focus on models relying on a standard ResNet-50 architecture for a controlled comparison in the main figure. VOneResNet features a hand-picked V1-like early layer and other refinements. The action recognition network also has a unique architecture. In the updated manuscript, we added statements referring to both networks in main text Section 4.1 first paragraph, alongside the other models, pointing to Appendix A.2 Figs. 8 & 9.

---

> > > ### Author Response · Authors · 2022-11-19
> > > **Initial response to reviewer JB2C (part 3 of 3)**
> > >
> > >
> > > >**QUALITY (MAJOR)**
> > >
> > > >Figure 4 claims that RS fits brain data better than adversarial training: do the baseline ImageNet accuracies of the RS and the adversarially trained models match? If not, can you provide an accuracy-matched comparison? Given that the ImageNet accuracy of the chosen adversarially trained models is very low, it is not surprising that they're also bad at fitting brain data
> > >
> > > In the updated Figure 4, RS no longer outperforms AT in accounting for V1 responses. We realized that the normalization we used for the AT model input was not matched to RS which affected the V1 result. We apologize for not having noticed this technical issue sooner. In response to the reviewer question about differences in ImageNet accuracy, we find that the robust networks are generally quite low accuracy-wise but that this is less of a concern for fitting V1 data. Indeed, ImageNet accuracy is more relevant for driving up fits of high-level brain area inferior temporal cortex (IT) as is well documented and we mention this caveat for those fits by saying “in part presumably because robust models have lower object classification performance which is known to drive fits in higher brain areas like V4 and IT supporting object recognition.”
> > >
> > > However, the manuscript is more concerned with early visual codes where the high-performing baseline ResNet-50 is particularly poor at predicting V1 responses despite its success in predicting IT neural responses. Future work will be needed to translate robust models to explaining IT data. This is a known deficiency of these networks relative to standard models as the reviewer alluded.
> > >
> > > >**QUALITY (MINOR)**
> > >
> > > >Movie interpolation example (Figure 3 left): I would suggest to show more examples in the appendix
> > >
> > > We have included three more examples in Appendix A.4.4, Figure 13. Please note that eleven movies total were used for the analysis (A.4.1, Figure 12 contains all frames from these movies) as these are the identical movies from the source study on perceptual straightening.
> > >
> > > >**NOVELTY (MINOR)**
> > >
> > > >In terms of the neural fits (e.g. Figure 4), it would help to explain where data is simply taken from brain-score.net and where it is computed/evaluated by the authors.
> > >
> > > In Section 4.2, we more explicitly state that we use brain-score for all neural comparisons throughout the manuscript.
> > >
> > > >**REPRODUCIBILITY**
> > >
> > > >Very poor - no code submitted, not enough details. For instance: How is the RS model trained? What is its ImageNet accuracy? Which hyperparameters, data augmentation, or even which deep learning library were used during training? I would strongly suggest to improve the reproducibility by providing more details in the paper itself, alongside submitting code if at all possible, and if this is not possible to explain why.
> > >
> > > We have included a statement on code & data availability in Appendix A.6. Also, under the section Methods, subsection 3.2 we explain how RS models were trained:
> > >
> > > “smoothing (Lecuyer et al., 2018; Cohen et al., 2019), a supervised network is trained but in the face of Gaussian noise added to the input space as the base classifier before performing a probabilistic inference.”
> > >
> > > The ImageNet accuracy for all the models were included in Table 1. For the details on training each model we provided the reference next to the model accuracy in Table 1. However, to explicitly state that the models that were used were exactly the same checkpoints, we added a sentence to the Table 1 caption.

---

> > ### Comment · Reviewer_JB2C · 2022-11-22
> > **Reviewer response to rebuttal**
> >
> > I would like to thank the authors for taking the time to take my feedback into account and write a comprehensive rebuttal, reflected by changes in the paper. I am happy with the changes; my concerns have been resolved. In particular, I appreciate the additional context regarding "brain-like" and the updated Figure 4. Two small observations for a potential camera ready version: some reference formatting issues persist, usually citations that refer to a specific paper by Author (YYYY) should be cited using \cite{} while other papers that are not referred to directly should be cited using \citep{} (Author, YYYY). Regarding reproduciblity, it's great that the authors now state code will be provided; that said it would be even better to submit it directly alongside the paper/rebuttal using e.g. an anonymous github link.
> >
> > In response to the author's changes, I am increasing my score (6 -> 8). Congrats again on a very neat paper!

---

### Official Review · Reviewer_J1tD · 2022-10-24

**Confidence:** 5
**Correctness:** 3
**Technical Novelty And Significance:** 1
**Empirical Novelty And Significance:** 4
**Recommendation:** 8

**Clarity, Quality, Novelty And Reproducibility:**

Clarity:
Overall the writing is clear.

Be careful with the claim of "perceptual plausibility" as it is only supported by SSIM and not by any perceptual data.

In section 3.3 the notion of expansion is introduced. As a reader it is not clear what this notion is intuitively, nor why it is supposed to be an interesting metric to control for later in the result section. It is somehow stated in section 4.2 but this would be great to have it written differently when the notion of expansion is introduced. How does this notion of expansion interact with curvature ?

A lot of citation should be in parenthesis (or use reference numbers ?). There is sometimes a missing space between a citation and the previous word. This is also the case for some references to figures. Equations in 3.3 : cos is a MathOperator in latex + large parentheses must be used.

Figure in the appendix are not referred to in the main text so they are somehow useless for the reader. An additional figure showing more example of interpolation between frames would be great in the appendix.
Text size in the figure should be almost as large as the main text.


Quality:
All sections are sufficiently good except for the discussion. There are several related questions like : (i) How perceptual movie straightening could affect image segmentation ? (ii) Is perceptual straightening still valid for stationary dynamical textures or are these textures already straight enough ? How perceptual straightening relates to stationarity ? (iii) What are the limitations of the perceptual straightening idea (flash-lag effect ?) ?


Novelty:
incremental but this is a very interesting step !

Reproducibility:
The authors do not state if their code will be released online

**Strength And Weaknesses:**

Strength:
- leverage behavioral and biological vision studies to question deep neural networks,
- relate robustness to input noise to perceptual straightening of movies,
- surprising observation : networks trained robustly on static images reproduces an observed dynamical feature of perception
- compare two types of robustness (random smoothing, adversarial training).

Weaknesses:
- incremental work
- results are not sufficiently discussed with the visual perception literature

**Summary Of The Paper:**

The paper tackles the question of straightening of natural movies in deep neural networks. It starts from previous observations both in visual psychophysics and neurophysiology that the curvature of the internal representations (abstract perception, V1 neural activity) is lower than the one of the initial movie representations (evolution of pixels in space-time). Such a reduction in curvature is not observed in the feature space (ouputs at different layers) of deep neural network (resnets) with standard training on large image datasets.

The authors shows that robust training (adversarial and random smoothing) is sufficient to obtain a curvature reduction in the feature space. According to the authors, this observation stems from a notion of "invertibility" which is used to describe a neural network in which linear interpolation in the feature space lead to perceptually smooth interpolation in the image space. They show that interpolating between movie frames generate perceptually plausible movies (using SSIM). Finally, the authors also shows that such robustly trained (random smoothing) neural networks are better at explaining the variance of neurophysiological recordings in the primary visual cortex.


**Summary Of The Review:**

This a good empirical contribution for ICLR filling a gap between vision studies and artificial neural network. The connection between a static feature of NN (robustness) and a dynamic feature of perception is appealing.

I will recommend acceptance more strongly once the authors have accounted for my remarks (ie improving the discussion section and referring to supplementary figures + fixing typos).

---

> ### Author Response · Authors · 2022-11-19
> **Initial response to review by J1tD (Part 1 of 2)**
>
> We would like to thank the reviewer for their helpful feedback and constructive suggestions to improve the manuscript. We attempt to address each of these below and hope that these changes improve the clarity and quality of the manuscript.
>
> >**MAJOR QUESTIONS**
>
> >Be careful with the claim of "perceptual plausibility" as it is only supported by SSIM and not by any perceptual data.
>
> Thank you for mentioning this, it is an important point to address. In Section 1 - Introduction, we have revised summary point #2, removing “perceptual plausibility” and instead stating:
>
> “...produces synthetic frames similar to those of the original natural movie sequence in image space.”
>
> Later, in the Results Section 4.1, second paragraph, when we introduce SSIM, we clarify its choice stating:
>
> “_Structural Similarity Index Measure _(SSIM (Wang et al., 2004)), that utilizes intermediate-level statistics motivated from biological vision and putatively more related to some aspects of human perception than simple pixel space correspondence.”
>
> >In section 3.3 the notion of expansion is introduced. As a reader it is not clear what this notion is intuitively, nor why it is supposed to be an interesting metric to control for later in the result section.
>
> In Methods, we clarify that expansion is simply the radius of the representation relative to the radius either for pixels (Fig. 14) or for a baseline resnet-50 (Figs. 6 & 15). We have added a schematic to Figure 5 that illustrates the radius metric which quantifies the size of the sphere covering the movie representation.
>
> >It is somehow stated in section 4.2 but this would be great to have it written differently when the notion of expansion is introduced. How does this notion of expansion interact with curvature?
>
> The rationale for looking at the radius of the movie representation is that perhaps straightening is a byproduct of the fact that these networks invoke loss functions meant to constrain variance to a smaller radius ball, so we might expect to see some relationship between radius of the movie and straightening if indeed representational size is somehow involved. This was merely an empirical question at this juncture.
>
> To further elaborate the relationship between curvature and size of the movie representation for the reader, we have included A.5, Figure 14 which shows how these quantities co-vary across model layers. We do not see any obvious trend though there is a slight correlation. In other words, networks tend to produce contracted representations for movies relative to pixel space and that seems to generally occur whether the network has high or low straightening scores.
>
> >A lot of citation should be in parenthesis (or use reference numbers ?). There is sometimes a missing space between a citation and the previous word. This is also the case for some references to figures. Equations in 3.3 : cos is a MathOperator in latex + large parentheses must be used.
>
> Thanks for pointing these out. We have gone through the manuscript and cleaned up all citation formats. We also have fixed parentheses in Equation 3.3, including adding the equation number.
>
> >Figure in the appendix are not referred to in the main text so they are somehow useless for the reader. An additional figure showing more example of interpolation between frames would be great in the appendix. Text size in the figure should be almost as large as the main text.
>
> - We have added main text references pointing to each of the appendix figures.
>
> - We added more examples of interpolation in a new appendix figure, A.4.4, Figure 13. We thank the reviewers for this suggestion as it is useful to see the results of inversion across a few types of movies. We added tables that breakdown the straightening and invertibility scores by movie and model to give interested readers more information, A.4.2 & A.4.3.
>
> - We have made an effort to clean up the text size in the figures for a better match to the text to improve clarity.

---

> > ### Author Response · Authors · 2022-11-19
> > **Initial response to review by J1tD (Part 2 of 2)**
> >
> > >Quality: All sections are sufficiently good except for the discussion. There are several related questions like : (i) How perceptual movie straightening could affect image segmentation ? (ii) Is perceptual straightening still valid for stationary dynamical textures or are these textures already straight enough ? How perceptual straightening relates to stationarity ? (iii) What are the limitations of the perceptual straightening idea (flash-lag effect ?) ?
> >
> > These are all interesting avenues to consider for discussion. Thanks for raising these as we had not thought of these connections initially.
> >
> > (i) For the relationship of movie straightening with image segmentation, it’s conceivable that perceptual straightening in vision helps support/guide stitching of bottom-up segmentation masks across frames. However, given that we did not do work directly in this active research area on segmentation, we refrain from further speculation but expect that this could lead to future empirical work to adequately address the relationship.
> >
> > (ii) Yes, we find that perceptual straightening is valid for stationary textures. This is a great question, and we have included a table breaking out straightening scores for each of the 11 movies tested, some of which were fairly stationary textures (e.g., leaves, water)(see section A4.1-4). It’s quite impressive that there is straightening and good movie interpolation for these more stationary patterns suggesting that even local motion statistics benefit from models in this class. We include a statement about this when we first present the invertibility result and at the end of the Discussion.
> >
> > (iii) In the updated Discussion, we acknowledge a limitation of our work is that the link of representational straightening to actual visual behavior is unclear. Perhaps this underlying straightened code is used for predictions (extrapolation) or postdiction (interpolation), but these are all open questions for further work. The flash-lag effect raised by the reviewer is a great example of how complicated perceptual estimates can be, and this illusion has been the subject of a debate on prediction versus postdiction mechanisms in perception. We briefly mention this illusion and how it raises interesting issues about visual inference. Perceptual straightening is merely an inferred representational property.
> >
> > >Reproducibility: The authors do not state if their code will be released online
> >
> > Code will be made publicly released online, and we have included a statement on code & data availability in Appendix A.6.

---

### Official Review · Reviewer_ePuu · 2022-10-24

**Confidence:** 4
**Correctness:** 3
**Technical Novelty And Significance:** 4
**Empirical Novelty And Significance:** 3
**Recommendation:** 6

**Clarity, Quality, Novelty And Reproducibility:**

The majority of the paper is clear. The paper could benefit from more clarity regarding the relationship between robust training and representational straightening (see the summary of the review). The findings of the paper are novel.

**Strength And Weaknesses:**

$\textbf{Strengths}$:

The results of this paper shed light on an important, previously observed phenomenon: namely, natural movie sequences are straightened both in human perception and primate V1 which is beneficial for linear prediction. These empirical observations implied that perception and neuronal representations underlying it are optimized for prediction of natural visual steams. Therefore, both predictive training and natural movies are both necessary for learning straightened representations in ANNs. This paper, interestingly shows that neither of these two factors are necessary. Rather, natural, static images and robust training are sufficient for learning straightened representations in ANNs. In addition to measuring the curvature of latent representations in a trained ANNs, authors use different metrics and experiments to explore this phenomenon from different viewpoints, which make the findings even more convincing. On top of that, the paper presents a computational model for primate V1 that finally shows a significant improvement compared to the previous models.

$\textbf{Weaknesses}$:

Although the empirical results shown in the paper are quite convincing, the paper doesn’t provide an intuitive understanding of why robust training leads to representational straightening of movies. The ANNs don’t see any natural movies during training, and there is no direct (or indirect) intuitive relationship between noise robustness in pixel space and linearizing sequence of frames in latent space. In the absence of such an explanation in the paper, my own efforts for finding an explanation directed me toward a possible relationship between robust training of natural images and predictive training of natural movies. I elaborate below in my comments.



**Summary Of The Paper:**

This paper presents an interesting and important finding regarding representational straightening of natural movies in artificial neural networks (ANN). The authors showed that representational straightening, which was previously shown in human perception of natural movies and neuronal representations in primary visual cortex, doesn’t necessarily require predictive training of ANNs with movies, and can emerge in ANNs trained with robust supervised learning algorithms (image classification plus adversarial training or random smoothing).


**Summary Of The Review:**

$\textbf{Questions}$:

1- The volume of the hypersphere in the latent space seems to have a relationship with the representational curvature of movies, based on the results presented in the paper (figure 6). Can the authors show a more thourough comparison of curvature and expansion metrics across different models and different layers of the models? It’s possible that the measured curvatures are, in general, smaller in more contracted latent hyperspheres which would also explain the results shown in figure 6.
This would also suggest an explanation for the observed relationship between robust training and straightening: robust training pushes similar images (eg consecutive frames of a movie) close together in the latent space. This could be considered as an implicit predictive training for movies, which is not dissimilar to self-supervised learning algorithms such as SimCLR and contrastive predictive coding. I'm interested to hear what the authors think about this possibility.

2- The outperformance of the RS model in accounting for V1 responses is very interesting. Given the significant difference between RS and AT in explaining V1 responses, how could we know if the outperformance of RS is caused by robustness, and not specifics of the RS algorithm?

3- The inverted linear interpolation experiments are very successful in giving a more direct evidence for the straightened representations learned via robust training. However, the SSIM metric used to quantify the similarity of natural movies and synthetic sequences mainly relies on lower-order statistics of the sequences. One could imagine that perceptual straightening relies on more abstract representations, in addition to lower-order statistics. In an ideal situation with enough time for more experiments, human observers would be the best evalutors for sequence similarities. In the absence of that possibility, could the authors provide more generated samples in the appendix. Especially, samples where the first and the last frames are more visually different. Also, which layers of the ANNs were used for the linear interpolations? How would the results look like for the early, intermediate, and late layers (both the appearance of the sequences and the SSIM results)? The most straightened representations seem to be in the intermediate layers. Would we also get the highest SSIM with the intermediate layer interpolations?


$\textbf{Minor Comments}$:

1- I suggest that the authors check the citation formats throughout the manuscript. There are a number of cases where the citation format is not correct, or the reference’s authors names is attached to the word before with no space in between. For example, see the last two lines of page 3.

2- figure 2 right: why not all AT models are shown in the curvature plots?

3- figure 5 caption: “Positive correlation means the smaller the correlation,…” correlation doesn’t seem to be the right word here.

4- page 7 last paragraph: “Between different RS models tested on different input noise levels, RS L2 stands out…” which RS L2?  Based on table 1, all RS models are L2.

---

> ### Author Response · Authors · 2022-11-19
> **Initial response to review by ePuu (Part 1 of 2)**
>
> We would like to thank the reviewer for their helpful feedback and constructive suggestions to improve the manuscript. We attempt to address each of these below and hope that these changes improve the clarity and quality of the manuscript.
>
> >**MAJOR QUESTIONS**
>
> >1- The volume of the hypersphere in the latent space seems to have a relationship with the representational curvature of movies, based on the results presented in the paper (figure 6). Can the authors show a more thourough comparison of curvature and expansion metrics across different models and different layers of the models? It’s possible that the measured curvatures are, in general, smaller in more contracted latent hyperspheres which would also explain the results shown in figure 6.
>
>
> We thank the reviewer for this intriguing suggestion as to a possible underlying explanation of the straightening phenomenon. Unfortunately, we do not see results that are clearly consistent with the idea that contraction and straightening are related and thus did not report on it in our initial submission. However, we have added a supplemental figure (Section A.5, Figure 14) elaborating on curvature vs. expansion across layers and models as these are still useful to report. We see that robust models stratify from non-robust ones in terms of the straightening metric but not as clearly along the expansion metric. To the reviewer’s point, some contraction for natural movies is happening in general perhaps because robust loss functions tend to incentivize invariance but given the mild trend, this may not be the only driving factor in improving straightening.
>
> >2- The outperformance of the RS model in accounting for V1 responses is very interesting. Given the significant difference between RS and AT in explaining V1 responses, how could we know if the outperformance of RS is caused by robustness, and not specifics of the RS algorithm?
>
>
> In the updated figures, RS no longer outperforms AT in accounting for V1 responses. We realized that the normalization we used for the AT model input was not matched to RS which affected the V1 result. Ultimately, AT performance is now marginally better than RS along all aspects tested (straightening, invertibility, and V1 fits). We apologize for not having noticed this technical issue sooner but believe that this change is warranted and does not change the main scientific message that certain forms of robustness training are conducive to the representational straightening of natural movies.

---

> > ### Author Response · Authors · 2022-11-19
> > **Initial response to review by ePuu (Part 2 of 2)**
> >
> > >3- The inverted linear interpolation experiments are very successful in giving a more direct evidence for the straightened representations learned via robust training. However, the SSIM metric used to quantify the similarity of natural movies and synthetic sequences mainly relies on lower-order statistics of the sequences._ _One could imagine that perceptual straightening relies on more abstract representations, in addition to lower-order statistics. In an ideal situation with enough time for more experiments, human observers would be the best evaluators for sequence similarities.
> >
> > We agree with the reviewer that SSIM is not capturing all the statistics necessary for perceptual similarity. We have added a statement to the main text (Results Section 4.1, Paragraph 2) stating this caveat that short of human perceptual judgments, SSIM is chosen as an improvement over simple correspondence of images in pixel space.
> >
> > >In the absence of that possibility, could the authors provide more generated samples in the appendix. Especially, samples where the first and the last frames are more visually different.
> >
> > Thank you for this helpful suggestion. We have included more generated samples in the appendix, A.4.4, Figure 13. In A.4.1, Figure 12 we show all 11 frames from all movies to give the reader a better idea of the visual differences across the full extend of the movie.
> >
> > >Also, which layers of the ANNs were used for the linear interpolations? How would the results look like for the early, intermediate, and late layers (both the appearance of the sequences and the SSIM results)? The most straightened representations seem to be in the intermediate layers. Would we also get the highest SSIM with the intermediate layer interpolations?
> >
> > Yes, presumably the absolute SSIM score would be higher if inversion is done using early or intermediate layers instead of the output layer, partly because of higher straightening scores in these layers and/or partly because earlier layers are fewer transforms away from the input. Since output layer interpolation has to pass through earlier layers, this strategy for interpolation provides a summary picture of the network’s overall straightening ability. From a biological standpoint, we viewed this analysis as a proxy for straightening at the level of perception in humans. Here it makes sense to invert the most downstream layers and look at the plausibility of the interpolated movie as an operational test of perceptual straightening in the models.
> >
> > While the absolute SSIM score may go up by using earlier layers for inversion, we expect that the relative pattern observed – of SSIM improvement for robust versus non-robust models – should be largely preserved.
> >
> > **MINOR COMMENTS**
> >
> > >1- I suggest that the authors check the citation formats throughout the manuscript. There are a number of cases where the citation format is not correct, or the reference’s authors names is attached to the word before with no space in between.
> >
> > We have gone through the manuscript and cleaned up all citation formats. Thanks for pointing this out.
> >
> > >2- figure 2 right: why not all AT models are shown in the curvature plots?
> >
> > Good suggestion. We now show all AT models in the Figure 2 curvature plots (right panel).
> >
> > >3- figure 5 caption: “Positive correlation means the smaller the correlation,…” correlation doesn’t seem to be the right word here._
> >
> > We have revised the Figure 5 caption to more clearly state: “Positive correlation means the smaller the size of the bounding hyper-sphere in pixel space, the more the straightened representation over the layers of the model.”
> >
> > >4- page 7 last paragraph: “Between different RS models tested on different input noise levels, RS L2 stands out…” which RS L2? Based on table 1, all RS models are L2.
> >
> > We have updated from “L2 stands out” to “L2: sigma = 0.5 stands out.” Thanks for pointing out the need for clarification.

---

### Decision · Program_Chairs · 2023-01-20

**Decision:**

Accept: poster

**Justification For Why Not Higher Score:**

Although the established connection between adversarial training/random smoothing and perceptual straightening appears to be novel and interesting this remains the main contribution of this paper which is purely experimental as it lacks any kinds of theoretical justification for the observed results.

**Justification For Why Not Lower Score:**

There is unanimous support for this paper to be accepted.

**Metareview: Summary, Strengths And Weaknesses:**

The paper investigates the degree to which adversarial training "straightens" visual representations (a concept borrowed from biological vision). A connection is drawn between adversarial training/random smoothing and perceptual straightening, which the reviewers found to be quite interesting. At the same time, it was also noted that the novelty was somewhat limited because the work builds heavily on concepts already established in the literature. It was also noted that the paper fails to provide any kind of theoretical justification for the found connection between adversarial training/random smoothing and perceptual straightening.




**Note From Pc:**

if the above contains the word "oral" or "spotlight" please see: "oral" presentation means -> notable-top-5% and "spotlight" means -> notable-top-25%. As stated in our emails, we are disassociating presentation type from AC recommendations